# Metabolic Engineering and Synthetic Biology Approaches for the Heterologous Production of Aromatic Polyketides

**DOI:** 10.3390/ijms24108923

**Published:** 2023-05-18

**Authors:** Dongsoo Yang, Hyunmin Eun, Cindy Pricilia Surya Prabowo

**Affiliations:** 1Synthetic Biology and Enzyme Engineering Laboratory, Department of Chemical and Biological Engineering (BK21 Four), Korea University, Seoul 02481, Republic of Korea; 2Metabolic and Biomolecular Engineering National Research Laboratory, Department of Chemical and Biomolecular Engineering (BK21 Four), Korea Advanced Institute of Science and Technology (KAIST), Daejeon 34141, Republic of Korea

**Keywords:** aromatic polyketide, enzyme engineering, metabolic engineering, polyketide, polyketide synthase, synthetic biology

## Abstract

Polyketides are a diverse set of natural products with versatile applications as pharmaceuticals, nutraceuticals, and cosmetics, to name a few. Of several types of polyketides, aromatic polyketides comprising type II and III polyketides contain many chemicals important for human health such as antibiotics and anticancer agents. Most aromatic polyketides are produced from soil bacteria or plants, which are difficult to engineer and grow slowly in industrial settings. To this end, metabolic engineering and synthetic biology have been employed to efficiently engineer heterologous model microorganisms for enhanced production of important aromatic polyketides. In this review, we discuss the recent advancement in metabolic engineering and synthetic biology strategies for the production of type II and type III polyketides in model microorganisms. Future challenges and prospects of aromatic polyketide biosynthesis by synthetic biology and enzyme engineering approaches are also discussed.

## 1. Introduction

Polyketides are a diverse set of natural products which are widely known for their medicinal activities such as antibiotic, antifungal, antiviral, anticancer, and anti-immunosuppressant activities [1]. Polyketides, comprised of carbon skeletons with β-keto groups with various degrees of reduction, are produced by a series of Claisen condensation reactions of short-chain acyl-CoA molecules such as acetyl-CoA or malonyl-CoA. Enzymes involved in the biosynthesis of polyketide main carbon skeletons are collectively called polyketide synthases (PKSs). After the biosynthesis of the main carbon skeletons, they can be modified by reduction, cyclization, aromatization, and other tailoring reactions (e.g., oxygenation, methylation, and glycosylation), leading to the production of diverse polyketide products. The biosynthetic mechanisms of PKSs are important in that such differences characterize the different types of PKSs and their products.

Depending on the biosynthetic mechanisms, PKSs can be largely classified into three types (I, II, and III) [1]. Among the three types, type I PKSs are again classified into modular type I PKS and iterative type I PKS. Modular type I PKSs are mostly megadalton-sized enzymes comprised of several modules (which are again comprised of several enzymatic domains) responsible for each carbon chain elongation step. A representative example of type I polyketide is erythromycin, an antibiotic, which has been successfully produced by engineered *Escherichia coli* [2]. There also have been other examples of successful production of type I polyketides including epothilone [3] and a rifamycin precursor [4] by engineered *E. coli*. Iterative type I PKS, on the other hand, consists of smaller enzyme clusters of which a single assembly line is iteratively utilized for the synthesis of the main carbon skeleton [5]. Representative examples of polyketides produced by iterative type I PKSs are lovastatin (used to treat cardiovascular diseases) [6] and simvastatin (a semisynthetic chemical used to treat high cholesterol), which have been successfully produced by engineered yeast [7].

In contrast to type I PKS, type II PKS refers to a group of discrete enzymes which cooperatively synthesize the main carbon skeleton. The three essential enzymes responsible for the carbon chain biosynthesis are termed the minimal PKS, which comprises ketosynthase α (KS_α_), ketosynthase β (KS_β_), and acyl-carrier protein (ACP). The KS_α_ and the KS_β_ subunits form a heterodimer which is responsible for the carbon chain elongation. The two subunits form an amphipathic channel at their interface which plays an important role in the determination of the carbon chain length and initial cyclization reactions. In particular, as the KS_β_ subunit is primarily responsible for determining the carbon chain length, which is directly relevant to the number of malonyl-CoA molecules incorporated, it is also termed the ‘chain length factor (CLF)’ [8]. The biosynthetic mechanism of type III PKS is simpler than that of type II PKS in that it only requires a single ketosynthase, which forms a homodimer, and that it utilizes malonyl-CoA instead of malonyl-ACP. In addition, unlike type II PKS which uses short-chain aliphatic CoAs (e.g., acetyl-CoA, propionyl-CoA) as starter units, a number of type III PKSs accept aromatic CoAs (e.g., coumaroyl-CoA) as starter units [8]. Therefore, many type III PKSs are also part of the chalcone synthase (CHS) family. Since the products of type II and type III PKSs often contain multiple aromatic rings, they are also termed aromatic PKSs. Aromatic polyketides include many chemicals important for human health such as antibiotics (e.g., oxytetracycline, actinorhodin) and anticancer agents (e.g., doxorubicin) [9].

Most polyketides are produced from soil bacteria such as *Streptomyces* species or plants (mostly type III polyketides). Although these hosts are remarkable organisms with abilities to produce a diverse set of polyketides, the following difficulties hinder such hosts from being used in either laboratory or industrial settings. *Streptomyces* species are spore-forming and slow-growing bacteria, which are notoriously difficult to engineer due to the limited availability of genetic engineering toolkits [10,11]. In addition, *Streptomyces* species show complex gene expression profiles according to different culture conditions due to the existence of diverse sigma factors and complex gene regulatory systems. On top of this, their culture conditions are very complex and the high-cell-density culture of these bacteria is not amenable, making it difficult to achieve high productivity and titer. In addition, many ‘undomesticated’ *Streptomyces* species still cannot be cultured in laboratory conditions. Even in easily cultivable *Streptomyces* species, a significant number of cryptic biosynthetic gene clusters (BGCs) that are not expressed unless in particular culture conditions make the problem worse. Due to poorly developed engineering tools, commercially utilized *Streptomyces* strains producing polyketides have been mostly developed by random mutagenesis. Culturing plants is also problematic since the cultivation often takes several months, in contrast to the time it takes for bacterial culture which is often several days. Additionally, the land and labor required for plant culture largely exceed those of bacterial culture, making it uneconomical. Therefore, efficient production of valuable polyketides by model microorganisms such as *E. coli* and *Saccharomyces cerevisiae* with well-established engineering tools is required to achieve optimal titer, productivity, and yield [12]. These model microorganisms are also capable of high-cell-density culture and have well-curated genome-scale metabolic models. Because engineering these hosts is highly amenable, they are also useful for the derivatization of polyketides, enabling the production of ‘unnatural’ polyketides through heterologous expression of diverse tailoring enzymes. Represented by the complete production of erythromycin in *E. coli*, type I polyketides have been readily produced from engineered *E. coli* strains [2,13]. However, heterologous production of aromatic polyketides from model microorganisms had been hampered until recently due to difficulties in the functional expression of type II and type III PKSs in heterologous hosts. The recent development of metabolic engineering and synthetic biology approaches have enabled facile production of several aromatic polyketides in model microorganisms, providing an avenue for the construction of efficient microbial cell factories for the production of diverse and useful aromatic polyketides [14,15].

In this review, metabolic engineering and synthetic biology strategies for the production of important aromatic polyketides in model microorganisms are reviewed with accompanying examples. The aromatic polyketides produced from heterologous microorganisms that are mentioned in this review are provided in Table 1. The readers are guided to excellent reviews on the heterologous production of polyketides in general [16,17].

## 2. Strategies for the Heterologous Production of Type II Aromatic Polyketides

Due to difficulties in engineering the native aromatic polyketide producers (mainly the *Streptomyces* species), as described earlier in the introduction, heterologous production of aromatic polyketides in model microorganisms such as *E. coli* or *S. cerevisiae* has been sought. During the last two decades, many researchers have delved into the production of aromatic polyketides by engineered *E. coli*, as it is one of the best-studied organisms with many characteristics favorable for industrial applications [30]. Such characteristics include fast growth, capability for high-cell-density culture, and well-established engineering tools [15]. However, attempts to produce type II polyketides in *E. coli* had mostly failed due to the highly insoluble and inactive type II minimal PKSs which were observed primarily as inclusion bodies [17]. Cloning of large-sized BGCs with the size of several tens of kilobase pairs with high guanine-cytosine (GC) contents was also problematic.

One of the first studies that enabled bypassing the minimal PKS solubility problem was achieved by engineering a fungal PKS (Figure 1A) [16]. In this study, PKS4 (an iterative type I PKS) from *Gibberella fujikuroi* was dissected and the KS, malonyl-CoA:ACP transferase (MAT), and ACP domains were reassembled to form a synthetic PKS that could mimic a type II PKS [28]. The engineered fungal PKS exhibited cyclization regioselectivity that had not been observed from a fungal PKS but rather from a bacterial type II PKS. Introduction of bacterial tailoring enzymes in addition to the synthetic minimal PKS resulted in the production of C18 polyketides PK8 and SEK26 (3 mg/L by fed-batch culture) [28]. A new nonaketide (produced from nine CoA units) nonaSEK4 could also be produced by the different combinations of tailoring enzymes, proving the versatility of employing *E. coli* as a host for PKS engineering. However, such a strategy could only be applied to a limited number of aromatic polyketides and the efficiency of the synthetic PKS was still very low. In another study, one of the major bottlenecks identified for aromatic polyketide biosynthesis was the extremely low expression level of a gene (*oxyB*) encoding the CLF (Figure 1B) [26]. Overexpression of an alternative sigma factor σ^54^ led to enhanced *oxyB* transcript level, thus facilitating the functional expression of the minimal PKS, resulting in 2 mg/L of oxytetracycline production in an *E. coli* strain harboring the oxytetracycline biosynthetic genes. However, the minimal PKS solubility issue was still not resolved by applying such strategies.

Recently, a pioneering study led by Takano and colleagues reported the discovery of a minimal PKS from the Gram-negative bacterium *Photorhabdus luminescens* TT01, which was soluble in *E. coli* [19]. Using the MiBiG (Minimum Information about a Biosynthetic Gene Cluster) repository [31], type II minimal PKS candidates were selected (Figure 1C). Then, minimal PKSs that are phylogenetically close to *E. coli* FabF were identified, which are more likely to be functional in *E. coli* when introduced. The minimal PKS comprised of AntD (KS), AntE (CLF), and AntF (ACP) from *P. luminescens* enabled efficient production of an octaketide (produced from eight CoA units) carbon skeleton required for the complete biosynthesis of anthraquinones. Using plug-and-play modes of production by the introduction of various combinations of tailoring enzymes, in addition to the minimal PKS, 1,3,7-trihydroxyanthracene-9,10-dione (AQ-256) could be produced, along with non-natural aromatic polyketides, neomedicamycin, and neochaetomycin [31].

Employing the same minimal PKS from *P. luminescens*, a natural red colorant carminic acid could be produced by metabolically engineered *E. coli* (Figure 1D) [24]. Introduction of *P. luminescens antDEF* (encoding the type II minimal PKS) along with *antB* [encoding a phosphopantetheinyl transferase (PPTase)] and *antG* (encoding a CoA ligase) allowed efficient biosynthesis of the C16 carbon skeleton. Additional introduction of *zhuI* and *zhuJ*, both encoding cyclases, from *Streptomyces* sp. R1128 resulted in the production of a carminic acid precursor, flavokermesic acid, with the titer as high as 180.3 mg/L. However, since the enzymes responsible for the remaining steps (monooxygenation and *C*-glycosylation) towards carminic acid production have not yet been discovered or were not functional in heterologous microorganisms, biochemical reaction analysis was performed to select enzymes that were expected to perform the desired reactions. DnrF from *Streptomyces peucetius* was selected as the monooxygenase and GtCGT from *Gentiana triflora* was selected as the *C*-glycosyltransferase [24]. As the native substrates of these enzymes were not flavokermesic acid, their activities were not enough for carminic acid biosynthesis. To enhance the activities of these enzymes, homology modeling was first performed to predict the 3D structures of the enzymes. Then, docking simulation-assisted mutagenesis of the enzymes was performed to identify enzyme variants with enhanced activities. Such strategies led to the production of carminic acid directly from glucose [24]. The GtCGT variant developed in this study (GtCGT^V93Q/Y193F^) was also capable of *C*-glycosylating other aromatic polyketides as well, as tested for the production of aloesin, a skin-whitening agent found from *Aloe vera*.

Another recent breakthrough was made by Jiang and colleagues by the discovery of a minimal PKS that enabled efficient production of C20 and C24 aromatic polyketides in *E. coli* (Figure 1E) [21]. In this study, the *alpA* and *alpB* genes from *Streptomyces ambofaciens*, each encoding KS and CLF, capable of biosynthesis of C20 carbon skeleton were discovered. Interestingly, the two genes were translationally coupled, which is probably one of the reasons for the soluble expression of the AlpA (KS) and AlpB (CLF) enzymes in *E. coli*. However, the exact mechanism of how translational coupling contributed to the soluble expression of the minimal PKS is not clear. Co-expression of the GroES/EL chaperone further enhanced the solubility of the minimal PKS. One problem that still remained was that the PPTase responsible for the activation of AlpC (ACP) was unknown. Therefore, an alternative ACP, RavC from *Streptomyces ravidus*, was employed. RavC could be readily activated by *Bacillus subtilis* Sfp (PPTase), resulting in the successful production of a decaketide (produced from 10 CoA units) dehydrorabelomycin (25 mg/L) [21]. It was also reported that additional introduction of AlpS, a thioesterase, could notably increase the efficiency of the biosynthesis of decaketide carbon skeletons, resulting in 0.5 g/L of dehydrorabelomycin production [22]. The research team also showed that employing the minimal PKS whiE-ORFIII, -ORFIV, -ORFV from *Streptomyces coelicolor* resulted in the production of a dodecaketide (produced from 12 CoA units) TW95c [21]. 

The elucidation of three-dimensional structures of type II PKSs can offer valuable insights into the biosynthetic mechanisms of aromatic polyketides and can guide potential engineering strategies. For instance, the recent discovery of the octaketide synthase minimal PKS complex from *P. luminescens* indicated that a hexaketide is covalently linked to the heterodimer with the ACP [32]. Such observations provide crucial insights for engineering type II PKSs, either to enhance biosynthetic efficiency or to develop entirely new enzymes for the production of diverse aromatic polyketides. Moreover, software such as PKMiner [33] or antiSMASH [34] can be employed to discover type II PKSs suitable for the biosynthesis of desired or novel aromatic polyketides. Such recent advancements in the heterologous expression of type II PKSs will contribute to the industrial production of medicinal polyketides as well as novel non-natural polyketide derivatives.

## 3. Strategies for the Heterologous Production of Type III Aromatic Polyketides

Type III PKS has a simpler biosynthetic mechanism when compared with type II PKS, as the KS homodimer is responsible for the carbon skeleton production. Moreover, the KS homodimer is often soluble in heterologous microorganisms; ACP is not required since CoA units are directly incorporated. One notable aspect here is that some type III PKSs can produce identical carbon skeletons that can be also produced by type II PKSs, although their biosynthetic mechanisms are different. This is the case when the same starter unit and the same number of extender units (malonyl-CoA) are used by both types of PKS. For example, when octaketide synthase (OKS; a type III PKS) from *Aloe arborescens* was introduced into *E. coli*, C16 octaketide shunt products SEK4 and SEK4b were produced [35], which can be also produced by bacterial type II minimal PKS in *E. coli* [24]. The versatility of type III PKS lies in the fact that site-directed mutagenesis leads to the production of different lengths of carbon skeletons. For example, engineering pentaketide chromone synthase (PCS) led to the production of octaketides SEK4 and SEK4b [36] and a non-natural nonaketide naphthopyrone [37]. Furthermore, engineering OKS led to the production of a tetraketide 6-acetonyl-4-hydroxy-2-pyrone, a pentaketide 2,7-dihydroxy-5-methylchromone, a hexaketide 6-(2,4-dihydroxy-6-methylphenyl)-4-hydroxy-2-pyrone, and a heptaketide aloesone [38]. Engineered OKS was also reported to produce the decaketide SEK15 [35], and the dodecaketide TW95a [39]. This was possible because changing the amino acids near the active sites changes the volume of the cavity in which the condensation reactions occur for the carbon chain elongation.

Due to its high solubility and catalytic efficiency, OKS has been a particularly useful enzyme. One of the most notable examples is the production of a natural red colorant carminic acid in *Aspergillus nidulans* [40] and in *S. cerevisiae* [20] by employing OKS. Production of flavokermesic acid, a precursor of carminic acid, was also achieved in *E. coli* by the introduction of OKS [24]. Due to its capability to efficiently biosynthesize C16 carbon skeleton, OKS was also employed in *S. cerevisiae* for the production of dihydrokalafungin, a precursor of an antibiotic actinorhodin [23]. Although the complete biosynthesis of actinorhodin could not be achieved due to the low activity of the later enzymes, the OKS-based production platform has paved the way towards efficient production of C16 aromatic polyketides by employing type III PKS. As such, the broad host range in which OKS can be applied proves that it is a highly versatile enzyme platform for the biosynthesis of aromatic polyketides.

Type III PKSs are also called as ‘chalcone synthase-like enzymes’ as many of them accept aromatic starter units to produce phenylpropanoids [41,42]. Type III PKS is a versatile type of PKS since it can accept a diverse array of starter units and is easier to engineer. While the starter units for type II PKSs are often limited to short-chain aliphatic acyl-CoAs (e.g., acetyl-CoA, propionyl-CoA, malonyl-CoA), the starter units for type III PKSs include a variety of CoA units such as hexanoyl-CoA, *p*-coumaroyl-CoA, and N-methyl-Δ^1^-pyrrolinium. For example, employing a chalcone synthase (CHS) from *Petunia × hybrida* enabled the production of naringenin chalcone from *p*-coumaroyl-CoA and malonyl-CoA, which led to the production of a phenylpropanoid dihydroquercetin (taxifolin) and the flavonolignans silybin and isosilybin, which are the active compounds from *Silybum marianum* (milk thistle) [43]. Olivetolic acid is another good example, which is the precursor of pharmaceutically important cannabinoids. Olivetolic acid biosynthesis requires the condensation reactions of hexanoyl-CoA and three malonyl-CoA molecules. A tetraketide synthase (TKS) from *Cannabis sativa*, responsible for the condensation reactions, was introduced together with olivetolic acid cyclase (OAC) in *E. coli* [25] and *S. cerevisiae* [44] for the production of olivetolic acid from glucose. In *S. cerevisiae*, olivetolic acid produced from galactose was prenylated with geranyl pyrophosphate (GPP), followed by several modification steps for the production of cannabinoids including Δ^9^-tetrahydrocannabinolic acid and cannabidiolic acid [44]. To avoid the formation of shunt byproducts by employing TKS, a highly reducing type I iterative PKS was engineered to specifically produce hexanoyl-CoA and a nonreducing PKS was engineered to produce olivetolic acid and its derivatives from *A. nidulans* [45]. Biosynthesis of acridones, produced by the condensation of anthraniloyl-CoA or N-methylanthraniloyl-CoA with three malonyl-CoA units, from engineered *E. coli* was also reported [46].

Production of tropane alkaloids, which are medicinal natural products including cocaine and atropine that are used to treat neurological disorders, involves a noncanonical type III PKS (AbPYKS) from *Atropa belladonna*. AbPYKS catalyzes the biosynthesis of 4-(1-methyl-2-pyrrolidinyl)-3-oxobutanoic acid by the condensation of an unusual starter unit N-methyl-Δ^1^-pyrrolinium and two molecules of malonyl-CoA [47,48]. In *S. cerevisiae*, the long biosynthetic pathway for the production of tropane alkaloids was divided and distributed to intracellular compartments such as mitochondria, peroxisome, vacuole, endoplasmic reticulum, cytosol, or vacuolar membranes to enhance the pathway efficiency. Such effort led to the production of hyoscyamine and scopolamine from engineered yeast [49]. As discussed above, the versatility of type III PKS allows the production of not only aromatic polyketides but also other diverse chemicals with distinct characteristics.

## 4. Metabolic Engineering and Synthetic Biology Strategies for Enhancing the Production of Aromatic Polyketides

For efficient production of aromatic polyketides, sufficient supply of precursors is important (Figure 2A). To this end, metabolic engineering of heterologous hosts for enhanced production of precursor acyl-CoAs has been sought. The primarily important precursor is malonyl-CoA, which is exclusively used as the extender unit for aromatic polyketide biosynthesis. As malonyl-CoA is also an essential metabolite, cells often produce malonyl-CoA to a level just enough for cell survival. Acetyl-CoA carboxylase (ACC) is responsible for the rate-controlling step in malonyl-CoA biosynthesis, which catalyzes the formation of malonyl-CoA from acetyl-CoA [50]. Therefore, ACC is one of the most important enzymes for the supply of precursors for aromatic polyketide biosynthesis. In particular, ACC from *Corynebacterium glutamicum* enabled sufficient supply of malonyl-CoA, leading to enhanced production of malonyl-CoA-derived aromatic polyketides in *E. coli* including phloroglucinol [27], 6-methylsalicylic acid, aloesone [18], and carminic acid [24]. Overexpression of *E. coli* ACC also resulted in enhanced production of olivetolic acid [25]. In another study, comprehensive phylogenetic and experimental tests were performed to select the most efficient ACCs among the diverse set of ACC families, leading to the identification of a previously unnoticed ACC from *Salmonella enterica*. Overexpression of *S. enterica* ACC led to more efficient production of malonyl-CoA when compared with the previously reported *C. glutamicum* ACC [51]. For example, employing *S. enterica* ACC resulted in high-level production (1074 mg/L) of (2*S*)-naringenin from glucose and tyrosine [51]. Oleaginous microorganisms such as *Yarrowia lipolytica* can be particularly useful for the production of aromatic polyketides due to the high metabolic flux towards acetyl-CoA biosynthesis. For example, establishing a pyruvate bypass pathway and overexpression of *PEX10* related to β-oxidation resulted in high-level production (35.9 g/L) of triacetic acid lactone (TAL) from glucose [29].

To efficiently construct microbial strains capable of producing a high level of malonyl-CoA, several malonyl-CoA biosensors were developed (Figure 2B). A representative malonyl-CoA biosensor is based on utilizing the transcription factor FapR [52]. FapR binds to the *fapO* operator site to block the expression of a reporter gene, followed by detachment from the operator upon coupling with malonyl-CoA. An enzyme-based biosensor was also developed using a type III PKS RppA which is capable of converting five molecules of malonyl-CoA into a red pigment flaviolin [18]. Therefore, the strain with higher malonyl-CoA production capability can be identified by the deeper red color of the culture with the naked eye. To exploit the capability of this biosensor, an *E. coli* genome-scale synthetic sRNA library was introduced to the strain harboring RppA to screen knockdown targets that significantly enhanced malonyl-CoA production. Biosensor-assisted high-throughput screening was especially effective in the identification of non-obvious knockdown gene targets; knockdown of *pabA* encoding *p*-aminobenzoate synthetase resulted in significant enhancement of malonyl-CoA accumulation. Applying the screened knockdown targets, including *pabA,* for aromatic polyketide biosynthesis resulted in enhanced production of 6-methylsalicylic acid, aloesone, naringenin, and resveratrol [18].

Some aromatic polyketides, such as doxorubicin, require propionyl-CoA as the starter unit. However, the de novo production of propionyl-CoA from simple carbon sources without feeding propionate is not an easy task, as exemplified by the production of erythromycin in *E. coli* [5]. To produce propionyl-CoA from malonyl-CoA, a 3-hydroxypropionate cycle that consists of malonyl-CoA reductase (MCR), malonic semialdehyde reductase (MSR), 3-hydroxypropionyl-CoA synthase (3HPCS), 3-hydroxypropionyl-CoA dehydratase (3HPCD), and acryloyl-CoA reductase (ACR) was constructed [53]. Additional introduction of propionyl-CoA carboxylases resulted in the production of (2*S*)-methylmalonyl-CoA as well, which is a useful precursor for the biosynthesis of macrolide type I polyketides. 

Heterologous cofactors are often one of the major bottlenecks for the heterologous production of aromatic polyketides. For example, the F_420_ cofactor required for the production of tetracycline was replaced with F_0_, a synthetically available derivative of F_420_, for the production of tetracycline from anhydrotetracycline in *S. cerevisiae* [54]. Additionally, supplementary enzymes for the activation of inactive apo-enzymes into active holo-enzymes are often required. A representative enzyme of this kind is Sfp, a PPTase from *B. subtilis*, which activates apo-ACP into holo-ACP. For example, introduction of Sfp and AntB, a PPTase from *P. luminescens*, led to the production of a type II polyketide carminic acid in *E. coli* [24]. Increasing the levels of inherent cofactors (e.g., NADPH, ATP) is also important. For example, increasing the NADPH levels by the knockout of *pgi* (encoding glucose 6-phosphate isomerase) and *ppc* (encoding phosphoenolpyruvate carboxylase) resulted in increased production of (+)-catechin (754 mg/L) in *E. coli* [55]. As discussed above, metabolic engineering and synthetic biology approaches for enhancing the production of aromatic polyketides require interdisciplinary studies with multifaceted approaches. Once a functional PKS is secured that enables the biosynthesis of aromatic polyketides with a desired carbon chain length, metabolic engineering and synthetic biology can take an important role by providing a holistic view of the metabolism, from the genomic level to transcriptomic, proteomic, metabolomic, and fluxomic levels [56].

## 5. Conclusions and Future Perspectives

In this review, PKS engineering strategies for the efficient production of type II and III polyketides from heterologous model microorganisms were discussed. Additionally, representative metabolic engineering and synthetic biology strategies for more efficient production of aromatic polyketides were discussed. Such advancements in the tools and strategies for type II and III PKS engineering have opened a new avenue for the efficient production of useful aromatic polyketides from model microorganisms. Recently, the diversification of natural products by attaching diverse functional groups followed by modification of the products has allowed the production of diversified new chemicals that are new to nature. Development of enzyme variants capable of performing desired reactions would facilitate the production of such novel chemicals. In addition, it is expected that the recently elucidated crystal structures of representative type II minimal PKSs will significantly boost the advancement of the corresponding field by providing more efficient minimal PKS libraries for the production of desired products [32,57,58,59]. In addition, the rapidly advancing machine learning-assisted accurate prediction of proteins using AlphaFold [60] or RoseTTAFold [61] will allow better understanding of the complex polyketide biosynthesis machinery, thus allowing the creation of PKSs for the production of tailor-made aromatic polyketides with medicinal or industrial importance.

Although engineering *Streptomyces* species had been notoriously difficult, recent advancement in genome engineering and genome editing tools has allowed easier manipulation of actinomycetal genomes. The CRISPR-Cas9 system [62] and the CRISPR-Base Editing SysTem (CRISPR-BEST) [63] were developed for efficient deletion, insertion, and base editing of the genomes of *Streptomyces* species. With such tools at hand, *Streptomyces* model strains which are relatively easier to engineer (e.g., *Streptomyces coelicolor*, *Streptomyces venezuelae*, *Streptomyces albus* and *Streptomyces lividans*) have been employed for efficient heterologous production of polyketides. This was showcased by the recent example of high-level avermectin production (9.3 g/L) from engineered *Streptomyces avermitilis* [64]. Such *Streptomyces* model strains are being perceived as potential model hosts for industrial production of important aromatic polyketides.

It is also worthwhile to note that the polyketide biosynthetic mechanism resembles that of fatty acid biosynthesis [65]. A recent study reported the production of a polyketide 6-heptyl-4-hydroxypyran-2-one by engineering a fatty acid synthase from *Corynebacterium ammoniagenes* [66]. This was possible by designing a synthetic fatty acid biosynthesis route that nonreductively elongates the carbon skeleton. Reversely, engineering PKSs for the production of non-polyketide small chemicals, such as short-chain ketones, was also reported [67]. Such strategies have not yet been reported for the production of aromatic polyketides, but engineering the PKS machinery holds a great promise by virtue of the great versatility of the PKS system.

The increasing emergence of antibiotic-resistant super bacteria has become a serious threat to modern society. The situation is worse than most people imagine as big pharmaceutical companies are rather reluctant to develop new antibiotics. This is because the development of new antibiotics costs a lot of money but the drug becomes useless within a short period of time due to the emergence of new antibiotic-resistant bacterial strains. Since polyketides are one of the main sources of natural antibiotics, more innovative strategies to produce diversified polyketides are required. The recent development of biosynthetic pathway design software (e.g., BNICE) that uses retrobiosynthesis methods is expected to aid in the prediction of new pathways for the production of desired aromatic polyketides [68]. The de novo design of new enzymes can be performed together with such strategies to enable the production of novel polyketides with new medicinal activities. Taken together, synthetic biology and enzyme engineering approaches for the production of diverse aromatic polyketides and their derivatives will continue to contribute to the supply of useful pharmaceuticals and also chemicals of other industrial importance.

## Figures and Tables

**Figure 1 ijms-24-08923-f001:**
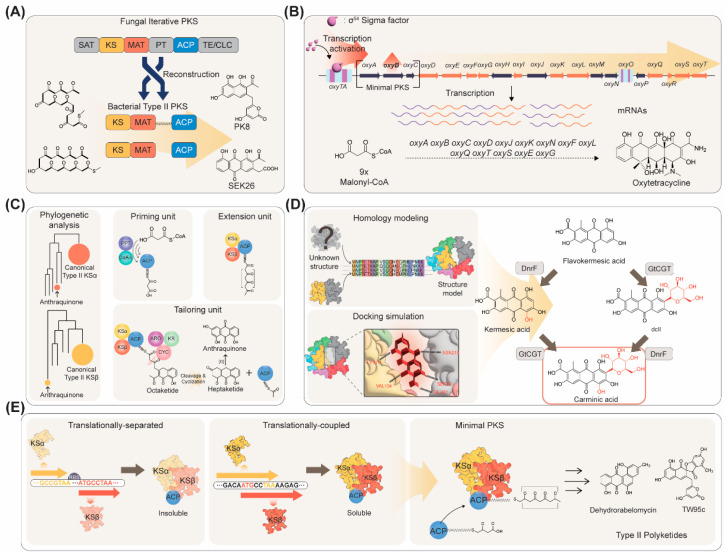
Minimal PKS engineering strategies for the production of aromatic polyketides. (**A**) Reconstruction of a fungal type II PKS into a synthetic PKS in bacteria for producing aromatic polyketides. (**B**) Overexpression of sigma factor σ^54^ to enhance *oxyB* transcript level, followed by increased production of oxytetracycline in *E. coli*. (**C**) Phylogenetic analysis-assisted plug-and-play modes of minimal PKS construction for aromatic polyketide production. (**D**) Combinatorial application of homology modeling and docking simulation of bottleneck enzymes to construct a novel carminic acid biosynthetic pathway in *E. coli*. (**E**) Translational-coupling of *alpA* and *alpB* genes to produce soluble forms of minimal PKS followed by increased production of C20 and C24 polyketides.

**Figure 2 ijms-24-08923-f002:**
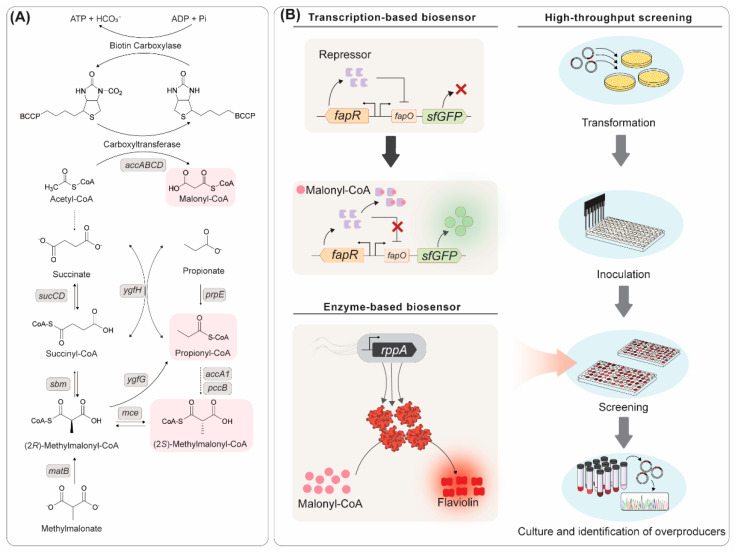
Metabolic engineering strategies for the production of aromatic polyketides. (**A**) Metabolic pathways to supply various precursors for aromatic polyketide production. (**B**) Transcription factor- and enzyme-based biosensors of malonyl-CoA to assist high-throughput screening of polyketide overproducers.

**Table 1 ijms-24-08923-t001:** The list of aromatic polyketides produced from recombinant microorganisms ^1^.

Product	Substrate	Concentration (mg/L)	Scale ^2^	Refs.
6-Methylsalicylic acid	Glycerol	440.3	2 L Fed-batch	[18]
Aloesone	Glucose	30.9	50 mL flask	[18]
AQ256	LB	2.5	6 L	[19]
Carminic acid	Glucose	7.58	2.5 L fed-batch	[20]
Dehydrorabelomycin	Tryptone, yeast extract, glycerol	500	250 mL flask	[21,22]
Dihydrokalafungin	Glucose	N/A	N/A	[23]
Flaviolin	Glucose	26	50 mL flask	[18]
Flavokermesic acid	Glucose	3660	2 L fed-batch	[24]
Neochaetomycin	Glucose	0.73	150 mL	[19]
Neomedicamycin	LB	1.04	4.8 L	[19]
Olivetolic acid	Glycerol	80	400 mL batch	[25]
Oxytetracycline	LB	2	25 mL flask	[26]
Phloroglucinol	LB	1280	2 mL batch	[27]
SEK26	Glucose	3	1 L Fed-batch	[28]
Triacetic acid lactone	Glucose	3590	3 L Batch	[29]

^1^ Phenylpropanoids and flavonoids are not included. ^2^ Working volumes are noted; detailed culture types are described in parentheses. N/A, not available

## Data Availability

Not applicable.

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
