# Peer review of "Metabolic Engineering and Synthetic Biology Approaches for the Heterologous Production of Aromatic Polyketides"

_ijms, 2023, doi:10.3390/ijms24108923_

Round 1

Reviewer 1 Report (Previous Reviewer 1)

No specific comments or suggestions. 

Author Response

Thank you for your time and effort in reviewing the paper.

Reviewer 2 Report (Previous Reviewer 2)

Congratulations

English is sufficient

Author Response

Thank you for your time and effort in reviewing the paper.

Reviewer 3 Report (New Reviewer)

The review article entitled "Metabolic engineering and synthetic biology approaches for the heterologous production of aromatic polyketides" submitted by Dongsoo Yang, Hyunmin Eun , Cindy Pricilia Surya Prabowo describes a review on the past and current trends and strategies on the metabolic engineering and heterologious production of aromatic ketides. Please see comments below:

1) The introduction and strategies for heterologous production of both type II and III aromatic ketides are well addressed with limited examples. 

2) In the section 2 of the manuscript, the authors mentioned 'due to the difficulties int he engineering the native polyketide producers', please describe what are the difficulties exist currently and list them how they can be overcome with heterologous production of aromatic polyketides.

Please address as minor revision for this manuscript.

Author Response

[Response] Thank you for your time and effort in reviewing the paper. We added further examples and discussions throughout the manuscript as follows:

Page 2:

“There also have been other examples on successful production of type I polyketides including epothilone [3] and a rifamycin precursor [4] by engineered E. coli.”

“Representative examples of polyketides produced by iterative type I PKSs are lovastatin (used to treat cardiovascular diseases) [6] and simvastatin (a semisynthetic chemical used to treat high cholesterol) which have been successfully produced by engineered yeast [7].”

Page 4-5:

“The elucidation of three-dimensional structures of type II PKSs can offer valuable insights into the biosynthetic mechanisms of aromatic polyketides and can guide potential engineering strategies. For instance, the recent discovery of the octaketide synthase minimal PKS complex from P. luminescens indicated that a hexaketide is covalently linked to the heterodimer with the ACP [26]. Such observations provide crucial insights for engineering type II PKSs, either to enhance biosynthetic efficiency or to develop entirely new enzymes for the production of diverse aromatic polyketides. Moreover, software like PKMiner [27] or antiSMASH [28] can be employed to discover type II PKSs suitable for the biosynthesis of desired or novel aromatic polyketides.”

Page 6:

“Biosynthesis of acridones, produced by the condensation of anthraniloyl-CoA or N-methylanthraniloyl-CoA with three malonyl-CoA units, from engineered E. coli was also reported [43].”

[Response] Thank you for the comment. We further discussed on the difficulties in the engineering the native polyketide producers and advantages of employing heterologous hosts in the introduction section as follows:

Page 2:

“Most polyketides are produced from soil bacteria such as Streptomyces species or plants (mostly type III polyketides). Although these hosts are remarkable organisms with abilities to produce diverse set of polyketides, the following difficulties hinder such hosts from being used in either laboratory or industrial settings. Streptomyces species are spore-forming and slow-growing bacteria, which are notoriously difficult to engineer due to the limited availability of genetic engineering toolkit [10, 11]. Also, Streptomyces species show complex gene expression profiles according to different culture conditions due to the existence of diverse sigma factors and complex gene regulatory systems. On top of this, their culture conditions are very complex and the high-cell-density culture of these bacteria is not amenable, making it difficult to achieve high productivity and titer. Also, many ‘undomesticated’ Streptomyces species still cannot be cultured in laboratory conditions.”

“These model microorganisms are also capable of high-cell-density culture and have well-curated genome-scale metabolic models. As engineering these hosts is highly amenable, they are also useful for the derivatization of polyketides, enabling the production of ‘unnatural’ polyketides through heterologous expression of diverse tailoring enzymes.”

In section 2, we revised the sentence as follows to indicate that such aspects are discussed in the introduction section.

Page 3:

“Due to the difficulties in engineering the native aromatic polyketide producers (mainly the Streptomyces species), as described earlier in the introduction, heterologous production of aromatic polyketides in model microorganisms such as E. coli or S. cerevisiae has been sought.”

The references were also changed accordingly.

This manuscript is a resubmission of an earlier submission. The following is a list of the peer review reports and author responses from that submission.

Round 1

Reviewer 1 Report

Nice review. No comments

Reviewer 2 Report

-insufficiently prepared, insufficient introduction, discussion and conclusions, review article insufficiently cited, not adequately discussed

- Keywords must be reordered; alphabetical order

-the paper should be checked according to the journal writing rules

-the fact of the matter is that the handling of the subject is complex and it seems that irrelevant information is given in most places